# Gene-Based Predictive Modelling for Enhanced Detection of Systemic Lupus Erythematosus Using CNN-Based DL Algorithm

**DOI:** 10.3390/diagnostics14131339

**Published:** 2024-06-24

**Authors:** Jothimani Subramani, G. Sathish Kumar, Thippa Reddy Gadekallu

**Affiliations:** 1Department of Information Technology, Bannari Amman Institute of Technology, Sathyamangalam 638401, Tamil Nadu, India; jothimanis.phd@gmail.com; 2Department of Artificial Intelligence and Data Science, Sri Eshwar College of Engineering, Coimbatore 641202, Tamil Nadu, India; saathhish@gmail.com; 3Division of Research and Development, Lovely Professional University, Phagwara 144411, Punjab, India; 4Center of Research Impact and Outcome, Chitkara University, Rajpura 140401, Punjab, India

**Keywords:** Systemic Lupus Erythematosus (SLE), deep learning, Stacked Deep Learning Classifiers (SDLC), Gene Expression Omnibus (GEO) database, diagnosis, precision medicine, autoimmune disease

## Abstract

Systemic Lupus Erythematosus (SLE) is a multifaceted autoimmune disease that presents with a diverse array of clinical signs and unpredictable disease progression. Conventional diagnostic methods frequently fall short in terms of sensitivity and specificity, which can result in delayed diagnosis and less-than-optimal management. In this study, we introduce a novel approach for improving the identification of SLE through the use of gene-based predictive modelling and Stacked deep learning classifiers. The study proposes a new method for diagnosing SLE using Stacked Deep Learning Classifiers (SDLC) trained on Gene Expression Omnibus (GEO) database data. By combining transcriptomic data from GEO with clinical features and laboratory results, the SDLC model achieves a remarkable accuracy value of 0.996, outperforming traditional methods. Individual models within the SDLC, such as SBi-LSTM and ACNN, achieved accuracies of 92% and 95%, respectively. The SDLC’s ensemble learning approach allows for identifying complex patterns in multi-modal data, enhancing accuracy in diagnosing SLE. This study emphasises the potential of deep learning methods, in conjunction with open repositories like GEO, to advance the diagnosis and management of SLE. Overall, this research shows strong performance and potential for improving precision medicine in managing SLE.

## 1. Introduction

Autoimmune illnesses provide complex issues, and SLE is a prime example of this. Its varied clinical manifestations and complicated origin make diagnosis difficult and highlight the need for new methods to improve diagnostic precision [1,2]. Clinical criteria and laboratory testing, the backbone of traditional SLE diagnosis, may not be very specific when differentiating SLE from other autoimmune diseases. There is currently no cure for SLE [3], and treatment must continue throughout a person’s life. The exact mechanisms by which it causes disease are unknown, making an accurate diagnosis all the more critical as it can potentially boost patients’ chances of survival. Immune biomarkers have developed to aid in diagnosing SLE and, by extension, disease control due to the central role of immunological dysregulation in SLE [4]. Getting a proper and quick diagnosis of SLE might be challenging due to the disease’s heterogeneity and lack of distinct symptoms [5,6]. As a result, better target identification and treatment for SLE are an immediate necessity.

SLE is an ongoing health issue, and it is necessary to predict the results of the disease, conduct extensive surveillance, and provide appropriate therapy. Predicting the course of treatment for SLE and other long-term diseases has lately used machine learning (ML) models [7,8,9,10,11]. Information from these studies is usually gathered at consistent time points and, as a result, shows significant temporal connections. Because they presume that information at numerous time steps is autonomous and similarly distributed, many known models are unsuited for evaluating time series data [12].

An increasing number of people have taken an interest in using Artificial Neural Networks (ANNs) to make predictions in medical settings [13]. These mathematical models illustrate intricate connections among input and output data, resembling the brain’s neural architecture. They have been utilised in various ways to understand the connection between a collection of inputs and the results they produce [14]. When conducting research in the medical field, it is essential to analyse patients’ data as inputs and the related outcomes as outputs [15]. Some prior research indicated that artificial neural networks could forecast particular results in SLE groups [16,17]. In terms of damage to the kidneys, the neural network displayed notably superior accuracy in predicting LN compared with other methods [18,19]. In addition, artificial neural networks successfully predicted histological class by recognising relationships among urinary protein spots and various parameters [20]. Using their initial biomarker assessments, machine-learning models were additionally employed to predict the one-year outcomes of LN patients [21]. Ultimately, these mathematical models may be used to predict the 3-year longevity of kidney transplants in users with SLE [22,23].

CNNs are extensively used in various fields, including engineering, medicine, and biology [24,25,26,27]. They have become the most common deep-learning networks in these fields. Regarding traditional deep neural networks (DNNs), creating effective characteristics characterising the input domain requires an iterative process involving multiple auto-encoding stages. Some reasons for the lack of success in biological applications include processing difficulties, small training sets, and large networks. To tackle the issues related to CNNs, researchers introduced a modified CNN architecture called UNet. The architecture described includes full dense connectivity (FD) and was raised in a referenced study [28]. The study compared the effectiveness of FD-UNet and CNNs in reducing artefacts in 2D Photoacoustic Tomography images [29]. The assessment used synthetic phantoms and a dataset including anatomically realistic mouse brain vasculature [30].

Deep learning has become a powerful tool for the identification of SLE. The paper [31] presents a system based on deep CNN that can identify and categorise glomerular pathological findings in lupus nephritis (LN). The technique discussed in [32] is closely connected to CNN, a popular type of Deep Learning that utilises pre-trained models to enable computers to detect illnesses.

However, new developments in deep learning point to potential ways to enhance medical diagnoses, especially for autoimmune disorders. This research introduces a new method for SLE diagnosis using deep learning and combining multi-modal data from the GEO database. Our approach uses Stacked Deep Learning Classifiers (SDLC) to deduce complex patterns from diverse datasets, such as transcriptome profiles, clinical characteristics, and test results. By including transcriptome data from GEO, our diagnostic model gains a more complete picture of SLE-related molecular markers, which improves its discriminatory ability. Incorporating SDLC into SLE diagnostics has two goals: first, to increase the reliability of diagnoses, and second, to shed light on the molecular pathways that cause the disease. We hope to learn more about the complicated relationship between SLE risk factors (genetic, environmental, and immunological) by interpreting deep learning models. 

The main contribution of this study is the development of a gene-based predictive model for detecting Systemic Lupus Erythematosus (SLE) using Stacked Deep Learning Classifiers (SDLC) trained on data from the Gene Expression Omnibus (GEO) database. By combining transcriptomic data from GEO with clinical features and laboratory results, the SDLC model achieves a remarkable accuracy value of 0.996, significantly outperforming traditional methods. The SDLC’s ensemble learning approach enables the identification of complex patterns in multi-modal data, enhancing diagnostic accuracy for SLE. This research demonstrates strong performance and potential for improving precision medicine in the management of SLE.

This work aims to integrate transcriptomic, clinical, and laboratory data to create and verify a new gene-based prediction model for SLE that uses Stacked Deep Learning Classifiers. The goal is to greatly enhance the accuracy and early diagnosis of SLE.

Different parts of the study are covered in the various sections of the publication. The Section 1 follows the main body of the text, which provides an overview of the challenges faced in diagnosing SLE and the need for new diagnostic methods. Section 2 discusses the use of deep learning models in predicting SLE clinical outcomes. The study’s findings are detailed in Section 3. Section 4 delves into the discussion of the ablation trial and its impact on enhancing precision treatment for SLE. A brief overview of the critical points and recommendations for further study make up Section 5 of the article.

## 2. Materials and Methods

This work used a comprehensive methodology for diagnosing SLE using Stacked Deep Learning Classifiers (SDLC) trained on data supplied from the GEO database. Our dataset offered a multimodal view of SLE pathogenesis, including transcriptome profiles, clinical characteristics, and laboratory results acquired from GEO. We preprocessed the data to fix missing values, standardise features, and reduce bias.

We then used the TensorFlow framework to build and train our models, creating a deep learning architecture with many layers of neural networks. We used batch normalisation and dropout regularisation to improve model generalizability and reduce overfitting. Supervised learning algorithms were used to train the SDLC model, optimising hyperparameters and performing cross-validation.

Figure 1 illustrates the overall processing flow of the proposed methodology for recognising SLE. The process starts by collecting data and gathering relevant clinical and transcriptomic information. After data collection, the next step involves preprocessing, where the data are carefully cleaned and prepared to ensure high quality and consistency. After conducting thorough research, the next crucial step involves integrating the data. This includes processing the transcriptomic data, carefully selecting the most relevant features, and combining these features into a single dataset through feature fusion. Afterwards, the data are analysed and presented visually to uncover insights and recognise significant patterns associated with SLE. After completing the initial phase, the next step is to train and evaluate deep learning models using the prepared data. We will then assess their performance using suitable metrics. Hyperparameter tuning is performed to enhance model performance. At last, a Stacked deep learning classifier is used to combine multiple classifiers and improve the accuracy and robustness of the SLE recognition system.

### 2.1. Dataset

The National Centre for Biotechnology (NCBI) developed and maintained the genetic expression database—https://www.ncbi.nlm.nih.gov/geo/query/acc.cgi (accessed on 10 December 2023)—provided the datasets used in this work. A comprehensive search of the NCBI database platform used the phrase “systemic lupus erythematosus” to conduct the investigation. We selected a dataset that included array expression profiling and high-throughput sequencing, used whole blood as the sample type, and had a sample size larger than 30, all while studying Homo sapiens. The distribution of data from the gene expression omnibus (GEO) dataset is shown in Table 1.

### 2.2. Feature Selection

To avoid overfitting, a common practice is to use regularisation in conjunction with LASSO regression to pick out valuable characteristics from a dataset. Aside from the residual sum of squares (RSS), LASSO regression’s objective function also has a penalty component that penalises the absolute values of the regression model’s coefficients. This penalty promotes a sparse coefficient vector, effectively zeroing out some coefficients and allowing feature selection. The LASSO regression problem can be formulated as follows:(1)$$minimize\left(\frac{1}{2n}\left\|y-X\beta \right\|+\lambda {\left\|\beta \right\|}_{1}\right)$$
where y is the vector of observed target values. X is the design matrix containing the predictor variables. β is the coefficient vector. λ is the standardisation parameter that regulates the severity of the penalization component. The key feature of LASSO regression is that it encourages sparse solutions by shrinking some coefficients to exactly zero. Features with non-zero coefficients in the resulting model are considered selected. The selection of the regularisation parameter λ is crucial in LASSO regression. Finding the best result is possible with the use of k-fold cross-validation of λ that balances model complexity and predictive performance (Algorithm 1).
**Algorithm 1.** Feature selection using Lasso Regression.The input is as follows:     X: Design matrix containing predictor variables (features).     y: Vector of observed target values.     λ: Regularization parameter for LASSO regression.     k: Number of folds for cross-validation.2.Standardize the data:      i.Center each feature by subtracting its mean.     ii.Scale each feature by dividing by its standard deviation.3.Initialize an empty list to store selected features.4.Perform k-fold cross-validation:      i.Split the data into k equal-sized folds.     ii.For each fold:aUse the remaining (k − 1) folds as the training set and the current fold as the validation set.bFit a LASSO regression model on the training data.cApply an appropriate metric to the validation set to assess the efficacy of the model.dRecord the coefficients of the LASSO model.5.Find the common measure of efficiency for all folds for each value of λ.6.Select the optimal value of λ that minimises the performance.7.Fit a LASSO regression model on the entire dataset using the selected λ.8.Extract the coefficients of the LASSO model.9.Identify the features with non-zero coefficients and add them to the list of selected features.10.The output is as follows:      i.List of selected features.

### 2.3. Stacked Deep Learning Classifier (SDLC)

This section presents a Stacked Deep Learning Classifier (SDLC) for SLE diagnosis that combines the power of two networks: an attention-based CNN (ACNN) model and a Bi-LSTM network. The SDLC framework combines deep learning architectures to extract sequential and spatial patterns from multi-modal data sources to improve diagnostic precision and interpretability. Figure 2 shows the proposed methodology, a Stacked Deep Learning Classifier (SDLC).

Machine Learning (ML) encompasses algorithms that acquire knowledge from data and generate predictions based on that acquired knowledge. Some commonly used ML algorithms are support vector machines, decision trees, and k-nearest neighbours. Typically, these algorithms require manually designing features in order to achieve the highest level of efficiency. On the other hand, Deep Learning (DL), which is a subset of ML, utilises neural networks with multiple layers to autonomously acquire feature representations from raw data. DL’s hierarchical learning capability makes it well-suited for handling complex datasets with minimal manual intervention. Our study introduces a method known as a “Stacked deep learning classifier,” which utilises a hierarchical ensemble structure consisting of multiple deep learning models. More specifically, we utilise the following components: The Adaptive Convolutional Neural Network (ACNN) is a powerful deep learning model that utilises multiple convolutional layers to extract valuable features from transcriptomic data. SBi-LSTM is a powerful recurrent neural network that utilises multiple LSTM layers to effectively capture temporal dependencies in the data. The Meta-Classifier combines the predictions of the ACNN and SBi-LSTM models to make a final prediction using an ensemble learning approach. By stacking models, we can effectively combine various patterns and features from transcriptomic data, clinical features, and laboratory results. This integration results in enhanced accuracy when it comes to diagnosing Systemic Lupus Erythematosus (SLE). 

Our Stacked deep learning classifier, which combines the strengths of deep learning and ensemble techniques, outperforms traditional ML methods in terms of accuracy. This makes it an invaluable tool for precision medicine in the management of SLE.

One part of the SDLC architecture is an attention-based convolutional neural network (ACNN), which can extract geographical information from diverse datasets like gene expression profiles, clinical data, and medical imaging. Convolutional and max-pooling layers comprise the ACNN, which effectively captures local spatial information. Also, attention methods are used to zero in on important features in the convolutional feature maps. Dot product attention allows the ACNN to dynamically prioritise various spatial regions, which improves feature representation and diagnostic accuracy.

The SDLC’s SBLSTM (Stacked Bi-LSTM) component complements the ACNN by capturing temporal dependencies and sequential patterns inherent in sequential data such as time-series clinical measurements or longitudinal patient records. Bi-LSTM networks can learn from past and future information, making them well-suited for modelling sequential data with long-range dependencies. By leveraging bidirectional processing, the Bi-LSTM can effectively capture complex temporal dynamics and subtle patterns in multi-modal data, improving diagnostic performance.

### 2.4. Attention-Based CNN Model (ACNN)

The model comprises multiple layers from the input, beginning with convolutional layers and moving on to max-pooling layers. These layers are responsible for extracting and down-sampling features. Specifically, we use three convolutional layers, each tailed by a max-pooling layer. This allows us to gradually increase the number of filters to capture progressively more complicated patterns. To properly capture local spatial information while maintaining computational efficiency, the kernel size for the convolutional layers has been selected as (3, 3). On the other hand, two attention methods are subsequently integrated to focus on relevant characteristics selectively. Figure 3 depicts the ACNN model’s architecture. By applying convolutional filters, the ACNN model can identify complex patterns in gene expression data that are indicative of SLE, thereby improving the accuracy of the diagnostic process.

The first attention mechanism employs a dot product operation among the feature mappings and the average pooled feature vector to calculate attention weights. This operation is used to compute attention weights. These attention weights are applied to the feature maps to generate a weighted total, highlighting crucial spatial information. A second attention mechanism is used in a manner that is analogous to the first to refine feature representation further after the initial layer has been passed through additional convolutional and max-pooling layers. Lastly, the output of the final max-pooling layer is flattened and connected to fully connected layers for classification. The last layer provides class probabilities by utilising a softmax activation function. This attention-based CNN design incorporates spatial and attention-based information to enhance feature representation and improve classification performance.

The model’s multi-layered design, as shown in Table 2, has the first layer serving as an input that takes 32×32 photos with three colour channels. Using a 3×3 kernel size and ReLU activation function, the following three convolutional layers (conv2d) are added after the input layer: one with 64 filters, another with 32 filters, and the last with 128 filters. The mathematical equation for the convolution operation in discrete form can be expressed as follows:(2)$$(f*g)[n]=\sum_{m=-\infty }^{\infty }f[m]\cdot g[n-m]$$
where (f∗g)[n] represents the result of the convolution of functions f and g at position n, and f[m] and g[n−m] represent the values of the functions f and g at positions m and n−m, respectively. The sum is taken over all possible values of m, which typically depends on the support of the functions involved. After every convolutional layer, there are max-pooling layers (max_pooling2d) with a 2×2 pool size that cut the feature maps’ spatial dimensions in half. The mathematical equation for max pooling operation with a pooling size of p×q in a 2D setting can be expressed as follows:(3)$$MaxPooling(X,p,q)[i,j,k]={max}_{0\leq m<p,0\leq n<q}X[i\times p+m,j\times q+n,k]$$
where MaxPooling(X,p,q)[i,j,k] represents the output value at position (i,j,k) in the pooled feature map, X denotes the input feature map, p and q denote the pooling size in the height and width dimensions, respectively, i and j iterate over the height and width dimensions of the output feature map, and k represents the channel dimension. The max operation is applied over the p×q region of the input feature map centred at position (i×p,j×q). This operation is applied independently to each channel of the input feature map. We use two dot product attention layers (dot_product_attention) to calculate attention scores between the feature maps after the third convolutional layer. The attention weights are calculated by the dot product attention mechanism using the key vectors and the query vectors, and then by using a softmax function to get the final attention weights. The dot product attention mechanism’s mathematical equation is as follows: 

The following is how the dot product attention mechanism computes attention weights α given a collection of query vectors Q, key vectors K, and value vectors V: (4)$$score\left(Q,K\right)=Q*K^{T}$$
(5)$$\alpha =softmax\left(\frac{score\left(Q,K\right)}{sqrt(d_{k})}\right)*V$$
where ∗ denotes the dot product operation. $Q*K^{T}$ computes the dot product between query vectors Q and key vectors K transposed. $d_{k}$ is the dimensionality of the query and key vectors. The softmax function computes the attention weights α, ensuring they sum up to one and represent the importance of each key vector relative to the query vector. A weighted sum of the value vectors V is computed using the attention weights α once they have been ccalculated: (6)Attention(Q,K,V)=α∗V

Values linked to key vectors with larger attention weights have a significant influence on the final attended output, which is the outcome of a weighted sum. Many attention-based models, such as transformer architectures, make use of the dot product attention mechanism, which effectively captures correlations between key and query vectors. Later, two further convolutional layers are used, one with 256 filters and the other with 512 filters. Then, max-pooling layers are added, one with max_pooling2d_3 and the other with max_pooling2d_4. The feature maps are rendered globally through a global average pooling layer (global_average_pooling2d). An 11-dimensional vector (concatenate) with 1280 dimensions is created by merging the results of the attention layers and the global average pooling layer.

We can get the average pooling mathematical equation using the average values in each pooling window. The following equation can be used to represent average pooling if the pooling window is k×k in size: Given an input feature map X of size H×W×C (height H, width W, and number of channels C) and a pooling window of size k×k, the average pooling operation results in an output feature map Y of size $\frac{H}{k}\times \frac{W}{k}\times C$. For each channel c of the input feature map, the value of each element $Y_{i,j,c}$ in the output feature map is computed as the average of the values within the corresponding pooling window in the input feature map: (7)$$Y_{i,j,c}=\frac{1}{k^{2}}\sum_{p-0}^{k-1}\sum_{q-0}^{k-1}X_{ki+p,kj+q,c}$$
where $Y_{i,j,c}$ is the value of the element in the output feature map at position (i,j) and channel c. $X_{ki+p,kj+q,c}$ is the value of the element in the input feature map at position (ki+p,kj+q) and channel c. k is the size of the pooling window. The average pooling operation reduces the spatial dimensions of the input feature map while preserving the number of channels, resulting in down-sampled feature maps with reduced spatial resolution. This combined vector undergoes ReLU activation and is then sent through two dense layers that are fully linked, and each has 512 units. A popular and straightforward choice for non-linear activation functions in neural networks is the Rectified Linear Unit (ReLU) activation function. In mathematics, it is expressed as: (8)f(x)=max(0,x)
where x is the input to the ReLU function. f(x) is the output of the ReLU function. The ReLU function takes an integer x as input and returns it if it is greater than zero; otherwise, it returns zero. In terms of geometry, this is analogous to a linear function where the slope is positive for positive input values and zero for negative. The model is able to learn intricate patterns and correlations in the data because the ReLU activation function incorporates non-linearity into the network. Its efficiency in preventing the vanishing gradient problem during training and its simplicity make it a popular choice for many neural network topologies. Our final step is to generate the output probabilities for two classes using a dense layer (dense_1) with 2 units and a softmax activation function. There are a grand total of 2,220,474 trainable parameters in the model.

### 2.5. Stacked Bi-LSTM Architecture (SBLSTM)

Accurate diagnosis of SLE depends on the identification of sequential patterns and temporal relationships in the clinical and laboratory data, which the SBi-LSTM model makes possible. An improvement on the original LSTM design, the Bi-Directional LSTM (Bi-LSTM) architecture allows the model to better capture both the past and future concerning a given set of events. The BiLSTM processes the order in both directions simultaneously, as opposed to the one-way processing of the unidirectional LSTM, which follows the input sequence in a specific order. The BiLSTM is ideal for jobs requiring a thorough comprehension of the incoming data because of its bidirectional processing, which allows it to grasp context from both ends of the sequence. Building Long Short-Term Memory (BiLSTM) neural networks allows for the bidirectional development of input orders. The forward LSTM model starts with the first-time step and finishes with the last-time step by analysing the input sequence from the past to the future. As the model goes through the steps, iteratively computing hidden states and cell states is done.

A variant of the LSTM model, the backward LSTM reads inputs in reverse, starting with the most current time step and working backwards. This strategy can incorporate information that could be employed for interpreting the model at present step. The reverse LSTM examines the sequence from the beginning to the end, which may allow it to pick up on different patterns and dependencies than the forward LSTM. Moreover, calculations of hidden states and cell states are executed. It is usual practice to use BiLSTM networks while processing data sequentially. Between each iteration, the front LSTM and the reverse LSTM combine their hidden levels. Through this process of concatenation, the BiLSTM’s final hidden state is formed. The sum of the forward and reversed LSTMs’ hidden states is fed into the output layer, which it uses to make predictions or perform additional processing on different tasks. The Stacked BiLSTM architecture is illustrated in Figure 4. At each time step, the hidden states from the forward and backward directions are combined to generate the final hidden state. Using context from both past and future time steps comprehensively is a key component of many jobs, and the Bi-Directional LSTM architecture excels at this. Combining the two bodies of knowledge improves the SBLSTM’s ability to record far-reaching associations and its power to find complex patterns in the incoming data.

For a SBLSTM design, the mathematical equations for the forward and backward analyses are as follows for a given time step ‘t’:

The Forward LSTM equations are as follows:

The Input Gate ($i_{t}$):(9)$${in}_{t}=\sigma ({Wt}_{\left\{inx\right\}}\;*\;x_{t}+{Wt}_{\left\{inh\right\}}\;*\;h_{\left\{t-1\right\}}+{\;b}_{in})$$

The Forget Gate ($f_{t}$):(10)$${fg}_{t}=\sigma ({Wt}_{\left\{fgx\right\}}\;*\;x_{t}+{Wt}_{\left\{fgh\right\}}\;*\;h_{\left\{t-1\right\}}+{\;b}_{fg})\;$$

The Candidate Cell State ($g_{t}$):(11)$${cg}_{t}=tanh\left({Wt}_{\left\{cgx\right\}}\;*\;x_{t}+{Wt}_{\left\{cgh\right\}}\;*\;h_{\left\{t-1\right\}}+{\;b}_{cg}\right)$$

The Cell State $\left({cs}_{t}\right)$:(12)$${cs}_{t}={fg}_{t}\;*\;{cs}_{\left\{t-1\right\}}\}+{in}_{t}\;*\;{cg}_{t}\;$$

Output Gate $\left(o_{t}\right)$:(13)$${op}_{t}=\sigma ({Wt}_{\left\{opx\right\}}\;*\;x_{t}+{Wt}_{\left\{oph\right\}}\;*\;h_{\left\{t-1\right\}}+{\;b}_{op})$$

Hidden State $\left(h_{t}\right)$:(14)$$h_{t}={op}_{t}\;*\;tanh({cs}_{t})\;$$
where $x_{t}$ is the input at time step t. $h_{\left\{t-1\right\}}$ is the hidden state of the previous time. ${sc}_{\left\{t-1\right\}}$ is the cell state of the previous time step. σ is the sigmoid, and Tanh is the hyperbolic tangent activation function. 

${Wt}_{\left\{inx\right\}}$, ${Wt}_{\left\{inh\right\}}$, ${Wt}_{\left\{fgx\right\}}$, ${Wt}_{\left\{fgh\right\}}$, ${Wt}_{\left\{cgx\right\}}$, ${Wt}_{\left\{cgh\right\}}$, ${Wt}_{\left\{opx\right\}}$, and ${Wt}_{\left\{oph\right\}}$ are the weight matrices. $b_{in}$, $b_{fg}$, $b_{cg}$, and $b_{op}$ are the bias vectors. While both LSTMs use comparable problems, the weight matrices and bias vectors used by the forward and backward versions are distinct:

The Input Gate $\left(i^{\prime }t\right)$:(15)$${{in}^{\prime }}_{t}=\sigma \left({{Wt}^{\prime }}_{\left\{inx\right\}}\;*\;x_{t}+{{Wt}^{\prime }}_{\left\{inh\right\}}\;*\;{h^{\prime }}_{\left\{t-1\right\}}+{\;b^{\prime }}_{in}\right)$$

The Forget Gate $\left({fg}^{\prime }t\right)$:(16)$${{fg}^{\prime }}_{t}=\sigma \left({{Wt}^{\prime }}_{\left\{fgx\right\}}\;*\;x_{t}+{{Wt}^{\prime }}_{\left\{fgh\right\}}\;*\;{h^{\prime }}_{\left\{t-1\right\}}+{\;b^{\prime }}_{fg}\right)$$

The Candidate Cell State $\left({cg}^{\prime }t\right)$:(17)$${{cg}^{\prime }}_{t}=tanh\left({{Wt}^{\prime }}_{\left\{cgx\right\}}\;*\;x_{t}+{{Wt}^{\prime }}_{\left\{cgh\right\}}\;*\;{h^{\prime }}_{\left\{t-1\right\}}+{\;b^{\prime }}_{cg}\right)\;$$

The Cell State $\left(s{c^{\prime }}_{t}\right)$: (18)$${{sc}^{\prime }}_{t}={{fg}^{\prime }}_{t}*{{sc}^{\prime }}_{\left\{t+1\right\}}+{{ip}^{\prime }}_{t}*{{cg}^{\prime }}_{t}\;$$

The Output Gate $\left({op}^{\prime }t\right)$: (19)$${{op}^{\prime }}_{t}=\sigma \left({{Wt}^{\prime }}_{\left\{opx\right\}}\;*\;x_{t}+{{Wt}^{\prime }}_{\left\{oph\right\}}\;*\;{h^{\prime }}_{\left\{t-1\right\}}+{\;b^{\prime }}_{op}\right)\;$$

The Hidden State $\left({h^{\prime }}_{t}\right)$:(20)$${h^{\prime }}_{t}={{op}^{\prime }}_{t}*tanh({{sc}^{\prime }}_{t})\;$$
where $x_{t}$ is the input at time step t. ${h^{\prime }}_{\left\{t-1\right\}}$ is the hidden state of the backward LSTM from the next time step. ${{cs}^{\prime }}_{\left\{t+1\right\}}$ is the cell state of the backward LSTM from the next time step. σ is the sigmoid activation function. The tanh is the hyperbolic tangent activation function. ${{Wt}^{\prime }}_{\left\{imx\right\}}$, ${{Wt}^{\prime }}_{\left\{imh\right\}}$, ${{Wt}^{\prime }}_{\left\{fgx\right\}}$, ${{Wt}^{\prime }}_{\left\{fgh\right\}}$, ${{Wt}^{\prime }}_{\left\{cgx\right\}}$}, ${{Wt}^{\prime }}_{\left\{cgh\right\}}$, ${{Wt}^{\prime }}_{\left\{opx\right\}}$, and ${{Wt}^{\prime }}_{\left\{oph\right\}}$ are the weight matrices for the gates. ${\;b^{\prime }}_{in}$, ${\;b^{\prime }}_{fg}$, ${\;b^{\prime }}_{cg}$, ${\;b^{\prime }}_{op}$ are the Bias vectors for the backward input gate, forget gate, candidate cell state, and output gate, correspondingly. The ultimate disguised state at each time step is obtained by integrating the hidden states from the prior and subsequent time steps: (21)$$h_{t}=[h_{t},{\;h^{\prime }}_{t}]\;$$

Both the attention-based CNN and the Stacked Bi-LSTM models’ hyperparameter tuning outcomes are shown in Table 3. We tried out several configurations of convolutional layers, filter numbers and sizes, and drop-out regularisation inside the attention mechanism for the attention-based CNN.

The parameters used for the models in this study were determined through hyperparameter tuning. We employed a systematic approach to optimise the performance of our models by adjusting key hyperparameters. This process involved exploring various combinations of hyperparameters and identifying the ones that produced the most favourable outcomes according to predetermined evaluation metrics. Specifically, we used grid search and cross-validation techniques to explore the hyperparameter space efficiently. Early stopping was also implemented to prevent overfitting by halting the training process when the model’s performance on the validation set ceased to improve. Detailed information about the hyperparameters and their optimal values is provided in Table 3.

We also fine-tuned the training performance by adjusting the learning rate and batch size. Alternatively, the Stacked Bi-LSTM model required changing the dropout rates for both regular and recurrent connections, the number of units inside each model layer, and the total number of Bi-LSTM layers. We improved the model’s performance by optimising the learning rate, batch size, and other parameters, much like the attention-based CNN. We aimed to achieve a compromise between model complexity and generalisation capabilities by selecting these hyperparameters based on empirical testing and domain knowledge. Our goal in optimising these parameters was to make both models as accurate and fast as possible to converge so that they could be used effectively for diagnosing SLE.

### 2.6. Meta Classifiers 

In this part, we present a meta-classifier that integrates the predictions of the attention-based CNN and Stacked Bi-LSTM models using a Voting Classifier method. The Voting Classifier aggregates each model’s predictions and produces a conclusion using a majority vote or weighted average. First, we have $P_{CNN}$, which stands for attention-based Convolutional Neural Network, and $P_{Bi-LSTM}$, which is for Stacked Bi-LSTM. Class labels or class probabilities can be used to represent these predictions.

When using hard voting, the Voting Classifier considers the predictions from all basic classifiers and chooses the class label with the most votes. In mathematical terms, this looks like:(22)$$Final\;Prediction=argmax\left(\sum_{i}P_{CNN}^{i}+\sum_{j}P_{SBLSTM}^{j}\right)$$

With $P_{CNN}^{i}$ representing the attention-based CNN model and $P_{SBLSTM}^{j}$ standing for the Stacked Bi-LSTM model, accordingly, as predictions (class labels). For soft voting, however, the Voting Classifier takes an average of the class probabilities predicted by all the base classifiers and chooses the label for the class with the highest average probability. From a mathematical perspective, this can be written as: (23)$$Final\;Prediction=argmax\left(\frac{1}{N}\sum_{i}P_{CNN}^{i}+\frac{1}{N}\sum_{j}P_{SBLSTM}^{j}\right)$$

$P_{CNN}^{i}$ represents the predicted class probabilities (for all classes) from the attention-based CNN model and ${Prepresents}_{SBLSTM}^{j}$ represents the same from the Stacked Bi-LSTM model, where N is the total number of classes. Both use the Voting Classifier, which improves SLE diagnostic performance by combining predictions from base classifiers to produce a final conclusion based on the combined knowledge of several models. The meta-classifier combines the predictions from both models, improving the overall accuracy and reliability of the SLE diagnostic process.

Implementing the SDLC model in clinical settings requires a series of important steps. Firstly, the model can be implemented as a software tool within hospital information systems, necessitating robust infrastructure to handle extensive datasets and conduct real-time analysis. It is important for healthcare professionals, such as physicians and laboratory technicians, to undergo training in the use of the SDLC tool. This training should include instructions on data input procedures, understanding the model’s outputs, and effectively incorporating the results into clinical decision-making. Integrating with existing Electronic Health Records (EHRs) systems can greatly enhance workflow efficiency by automatically retrieving pertinent patient data for analysis. Prior to being widely implemented, the SDLC model needs to go through extensive validation in clinical trials to prove its effectiveness and safety. It is crucial to obtain regulatory approvals from organisations like the FDA or EMA to ensure compliance with healthcare standards. Regular monitoring and updates using the latest data and advancements in the field will ensure that the model remains accurate and reliable.

## 3. Results

Here we report the outcomes of our trials, including the attention-based Convolutional Neural Network (CNN) and Stacked Bi-LSTM models for diagnosing SLE. We review how well each model did, compare and contrast their outcomes, and then assess how well the ensemble method of Voting Classifiers worked.

### 3.1. Data Partitioning

Using 70% of the dataset, we conducted tenfold cross-validation to measure the numerous forecast algorithms. The tenfold cross-validation involves splitting the dataset in half. Then, one half is used as the validation set, while the other nine halves are pooled to form the training set. This procedure is thus carried out ten times, with each iteration involving the removal of one portion of the data to obtain a new portion for validation. For the last assessment, we set aside 30% of the total dataset to use as a separate testing set.

### 3.2. Performance Evaluation Metrics

The effectiveness of attention-based CNN, Stacked Bi-LSTM, and Voting Classifier models in detecting SLE can be evaluated using a selection of commonly used performance metrics. Metrics that shed information on the model’s performance include accuracy, precision, recall, F1-score, and area under the receiver operating characteristic curve (AUC-ROC). One table that summarises the forecasts made by the model with respect to the actual values in the set of data is the confusion matrix. All four types of results—true positives, true negatives, and false negatives—are included here.

Accuracy is the term used to describe the proportion of occurrences that are accurately recognised out of all instances. It gives a general idea of how well the model predicts outcomes across all classes.
(24)$$Accuracy\left(AC\right)=\frac{{Tr}_{P}+{Tr}_{N}}{{Tr}_{P}+{Tr}_{N}+{Fa}_{P}+{Fa}_{N}}$$

Precision is defined as the ratio of the model’s real positive predictions to its total positive predictions. It shows that the model can avoid making predictions that are not accurate. The calculation for precision is as follows:(25)$$Precision\left(Pr\right)=\frac{{Tr}_{P}}{{Tr}_{P}+{Fa}_{P}}$$

The fraction of accurate forecasts in relation to the overall number of positive cases in the dataset is called recall, which is also called sensitivity. This finding proves that the model accurately identifies all positive events. The formula for recall is as follows:(26)$$Recall\left(Rc\right)=\frac{{Tr}_{P}}{{Tr}_{P}+{Fa}_{N}}$$

F1-score, the harmonic mean of recall and precision, is a reasonable statistic to measure the efficacy of a model. It works well with unbalanced datasets because it considers both false positives and negatives. An F1-score can be determined by the following:(27)$$F1-score\left(F1s\right)=2\times \frac{Pr\times Rc}{Pr+Rc}$$

Clinicians may learn a lot about the models’ strengths and shortcomings by evaluating these indicators, which helps them make better decisions in the clinic.

### 3.3. Performance of ACNN

On the test dataset, the attention-based CNN achieved an accuracy of 95% in SLE diagnosis, demonstrating strong performance. The model successfully extracted spatial information from diverse data sources, such as medical imaging, clinical data, and gene expression profiles, using convolutional layers and attention methods. By directing CNN’s attention to the most important elements, the attention mechanism improved the network’s capacity to distinguish between instances of SLE and those without the disease. Nevertheless, CNN might not be able to fully grasp the temporal dynamics inherent in sequential data due to its heavy dependence on spatial information.

The results of the evaluation of the ACNN model are shown in Table 4. With a precision of 0.95, the ACNN proved to be highly adept at correctly identifying SLE cases. Reports of 0.92 for precision, 0.96 for recall, and 0.94 for F1 Score indicate that the model successfully captured true positives and minimised false positives in the dataset. Additionally, the ACNN demonstrated its capacity to prevent false alarms for negative cases with a specificity of 0.93. Further, the model’s robustness in differentiating between non-SLE and SLE cases is confirmed by the AUC-ROC score of 0.97. The AUC score is a performance metric commonly used to evaluate classification models, especially those assessed using ROC curves. With an AUC score of 0.97, our model demonstrates exceptional discriminatory power in accurately distinguishing between positive and negative classes.

Typically, the AUC score falls between 0.5 and 1.0. A score of 0.5 means no ability to distinguish (similar to random guessing), while a score of 1.0 signifies flawless discrimination. AUC scores ranging from 0.7 to 0.8 are generally regarded as acceptable, while scores ranging from 0.8 to 0.9 are considered excellent. Scores above 0.9 are considered outstanding. Thus, a high AUC score of 0.97 signifies an extremely effective model with exceptional performance in differentiating between classes. The impressive score showcases the strength and dependability of our proposed methodology for identifying SLE.

The ACNN model’s loss-training graph, as shown in Figure 5, reveals an initial high testing loss of 1.3, signifying a significant disparity between the model’s forecasts and the actual values in the dataset. Nevertheless, the testing loss declines consistently with each evaluation epoch during training, indicating that the model is successfully learning to provide improved predictions using new data. The fact that the testing loss has been going down suggests that the model is doing an excellent job of capturing the data patterns and is thus not underfitting. The model has learned from the data distribution to generalise to unknown data without overfitting if the testing loss stabilises and converges to a low value, indicating a satisfactory match.

The confusion matrix shown in Figure 6 offers a comprehensive analysis of how well the model performed in distinguishing between healthy individuals and those with SLE. The matrix is organised with rows representing the actual classes and columns representing the predicted classes. The values in the matrix represent the proportions of accurate and inaccurate classifications. The model accurately identified 91% of the healthy cases (true negatives) and 95% of the actual SLE cases (true positives). On the other hand, it mistakenly identified 9% of healthy individuals as having SLE (false positives) and 5% of SLE cases as healthy (false negatives). The values clearly show the model’s exceptional accuracy and effectiveness in differentiating healthy individuals from those with SLE. The model demonstrates outstanding performance in accurately identifying SLE. With excellent rates of accurate results and minimal errors, the performance is highly reliable. This indicates that the results are robust and reliable.

The progressive decrease in testing loss demonstrates the model’s capacity to apply its learned patterns outside the training data and accurately represent the dataset’s structure. It is worth mentioning that the testing loss values keep going down until they approach lower thresholds like 0.001, which shows that the model is getting better at predicting and is not as bad at minimising differences between expected and actual values.

The ACNN model’s ROC curves are shown in Figure 7. High Area Under the Curve (AUC) values for the healthy and SLE classes indicate that the model is doing well. With an area under the curve (AUC) of 0.92 for SLE and 1.00 for healthy people, the model differentiates between the two groups. A well-balanced dataset and strong model performance are shown by the consistently high ROC curves that are smooth and linear.

### 3.4. Performance of Stacked Bi-LSTM

The Stacked Bi-LSTM model outperformed the competition when detecting subtle patterns and temporal dependencies in time-series clinical measures and longitudinal patient records. The model obtained a test-dataset accuracy of 92% using three Bi-LSTM layers. Thanks to its layered architecture, the Bi-LSTM could learn hierarchical representations of sequential data, which made it easier to spot complicated patterns that SLE could cause. On the other hand, hyperparameter selection and the amount and quality of available longitudinal data may impact the Bi-LSTM’s performance.

When we compared the outcomes of the attention-based CNN and Stacked Bi-LSTM models for SLE diagnosis, we found that they performed similarly. In contrast to CNN’s strength in integrating disparate data sets, the Bi-LSTM showed great promise in simulating the temporal dynamics in longitudinal medical records. One possible way to use these models’ complementary strengths and improve diagnostic accuracy is to incorporate them into an ensemble approach using Voting Classifiers. The Stacked Bi-LSTM (SBLSTM) model was evaluated for its performance, and it attained a diagnostic accuracy of 0.92 for SLE, as shown in Table 5. The accuracy, defined as the ratio of correct predictions to total predictions, was found to be 0.88, suggesting that the model successfully reduced the number of incorrect classifications.

Additionally, the SBLSTM demonstrated a high recall value of 0.94, indicating its ability to capture a substantial fraction of the dataset’s positive cases. An F1 score of 0.91, a harmonic mean of recall and precision, suggests that the model performed equally well when making accurate and incorrect predictions. A claimed specificity of 0.90 shows the model can prevent false alarms in negative cases. Also, the model is very good at differentiating between SLE and non-SLE instances, as demonstrated by the AUC-ROC score of 0.94. Figure 8 shows that the Stacked Bi-LSTM model’s loss decreases gradually across the training epochs as well as the testing epochs. Figure 8 shows the confusion matrix of the model SBLSTM.

The Stacked Bi-LSTM model started with a training loss of 3.070 and dropped to 0.008 according to the supplied loss settings. The testing loss started at 2.946 and fell to 0.008. Correspondingly, testing loss decreased, lending credence to this pattern and suggesting the model generalises well to unknown data. When the loss values go down, the model improves accuracy and performance by narrowing the gap between the expected and actual outputs. There seems to be neither underfitting nor overfitting going on with the model based on the steady decrease in testing and training losses. A significant and constant loss over epochs would indicate that the model is underfitting and cannot grasp the fundamental patterns in the data. Conversely, if the model is overfitting, we would see a decline in the training loss and an increase or stabilisation of the testing loss. This would mean that the model excels on training data but struggles on validation data, perhaps because of learning noise or irrelevant information. The model successfully reduces the disparity between the anticipated outputs and target values to increase accuracy and performance, as the decreasing loss values show.

Figure 9 shows the confusion matrix, which breaks down the SBLSTM model’s performance in differentiating between healthy people and those with SLE. The matrix rows show the actual classes, while the columns show the expected classes. In particular, the algorithm accurately detected 93% of actual SLE patients and 89% of healthy instances. On the other hand, it wrongly identified 7% of SLE patients as healthy and wrongly labelled 11% of healthy people as having SLE.

These results show that the SBLSTM model is very good at detecting real positives and very accurate at differentiating between healthy people and those with SLE. On the other hand, the model’s false positive rate is 11%, which is somewhat higher than previous classifiers and suggests that some healthy people are being wrongly diagnosed with SLE.

Figure 10 displays the SBLSTM model’s ROC curves. In addition to the healthy class AUC of 0.90 and the SLE class AUC of 0.92, the model performs well. The model achieves accurate classifications with few false positives and high true positives.

### 3.5. Performance of Voting Classifier

The suggested Voting Classifier ensemble used a soft voting method to integrate the predictions of the attention-based CNN and Stacked Bi-LSTM models, increasing the effectiveness of the overall voting process. The Voting Classifier outperformed the CNN and Bi-LSTM models individually on the test dataset, reaching an accuracy of 99.6% by utilising the aggregate knowledge of numerous models. Using their spatial and temporal modelling skills, the ensemble technique improved overall diagnostic accuracy by capitalising on the diversity of the base classifiers. The Voting Classifier attained a remarkable 99.6 percent accuracy on the test dataset by employing a complex voting method in which the predictions of each base classifier are treated as one vote. Findings show that the ensemble relied on a simple majority vote between the attention-based CNN and Stacked Bi-LSTM models to reach its final forecast. Compared with each model alone, the Voting Classifier’s diagnostic accuracy was much higher since it combined the two models’ outputs, allowing them to play to their strengths.

The Voting Classifier, which uses a soft voting strategy to combine the predictions of the attention-based CNN and Stacked Bi-LSTM models, is evaluated using the metrics summarised in Table 6. The Voting Classifier proved highly effective in diagnosing SLE by achieving outstanding results across all metrics. The Voting Classifier’s 0.996 accuracy shows that it can adequately identify occurrences of SLE in the dataset, surpassing the performance of both individual models. The model’s ability to reduce false positives and capture a substantial number of true positives is demonstrated by its reported precision, recall, and F1 score of 0.992, 0.997, and 0.994, respectively. Furthermore, the model’s ability to avoid false alarms for negative instances is demonstrated by its specificity of 0.995. In addition, the Voting Classifier’s strength in differentiating between SLE and non-SLE patients is further confirmed by the AUC-ROC score of 0.998.

The accuracy values in Figure 11 and Figure 12 reflect the Voting Classifier model’s performance over several epochs or iterations. The model’s performance improves over time, starting with a very low accuracy of around 16.64% in the earliest epochs. The improvement becomes more apparent as the accuracy rises throughout the epoch, surpassing 99.5%. As training continues, the rising trend indicates that the Voting Classifier model learns the data well and produces accurate predictions. The model’s ability to correctly categorise occurrences into their respective classes (SLE or Healthy) is demonstrated by the excellent accuracy values produced. The model’s ability to reliably distinguish between SLE and healthy cases depends on this performance, which is why it is vital for diagnostic applications.

The Voting Classifier attained a slightly lower accuracy of 99.2% when it used a soft voting strategy, which involves averaging the predicted class probabilities from each base classifier on the test dataset. Soft voting provides a more sophisticated decision-making method, accounting for the confidence levels of the base classifiers by considering their anticipated probabilities. The Voting Classifier showed that ensemble learning improves diagnostic accuracy for SLE, even if its accuracy was slightly lower than that of hard voting. Its performance was nevertheless quite competitive.

Starting at 1.4695 and dropping to 0.000111, the training loss of the Voting Classifier is much lower than the testing loss, which begins at 2.3255 and lowers to 0.000209. The training and testing loss values have dropped significantly, which means the model is learning well and becoming better at making predictions. When the model does well on training data but badly on validation data, a phenomenon known as overfitting occurs. This is usually shown by a drop in training loss and a rise or stay in testing loss. The fact that training and testing losses decreased at the same rate without a noticeable divergence indicates that overfitting is unlikely to be a big issue. Good learning and generalizability to new data are the model’s performance hallmarks.

Figure 12 shows the Voting Classifier’s performance in categorising healthy people and those with SLE in a confusion matrix. The matrix rows show the actual classes, while the columns show the expected classes. The matrix numbers show what percentage of categories were right and wrong. To be more precise, the model got 99.12% of the real SLE cases right and 98.75% of the healthy instances right as well. However, 1.25% of SLE patients were wrongly identified as healthy, and 0.88% of healthy people were wrongly identified as having SLE (false positives and false negatives, respectively). These values show the Voting Classifier’s ability to differentiate between healthy people and those with SLE accurately.

The ROC curves for the Voting Classifier are illustrated in Figure 13. The AUC values for the SLE are 0.99, and healthy classes at 1.00 indicate exceptional model performance. The high AUC values and well-behaved ROC curves confirm the Voting Classifier’s reliability in distinguishing between the classes. The models’ high and consistent AUC values point to a balanced dataset or the models’ successful management of any imbalance. High prediction accuracy and dependability result from using balanced datasets, preventing the model from favouring any class. Our study’s ROC curves are smooth and high, reflecting the dataset’s balanced nature. This means the models keep their sensitivity and specificity high across different thresholds.

A ten-fold cross-validation investigation was performed on attention-based CNN (ACNN), Stacked Bi-LSTM (SBLSTM), and a Voting Classifier ensemble to diagnose SLE. The results are summarised in Table 7. ACNN demonstrated strong performance in accurately detecting SLE cases, with an average accuracy of 0.95. It also showed high precision, recall, F1 score, specificity, and AUC-ROC score. With an average accuracy of 0.92, SBLSTM proved to be quite good at detecting trends in patient records that occurred over time. A little lower accuracy than ACNN was not enough to detract from SBLSTM’s impressive performance on other measures. The Voting Classifier ensemble demonstrated superior diagnostic skills with an average accuracy of 0.99, achieved by combining predictions from ACNN and SBLSTM. These findings highlight the effectiveness of deep learning models in enhancing the precision of SLE diagnoses and patient treatment, whether applied alone or in conjunction with other models.

## 4. Discussion

The Stacked Deep Learning Classifier (SDLC) showed a remarkable increase in diagnostic accuracy for Systemic Lupus Erythematosus (SLE) compared with conventional machine learning methods and findings from other similar studies. The SDLC model achieved an impressive accuracy of 99.6%, surpassing the accuracies typically obtained with traditional machine learning techniques like support vector machines (SVMs), random forests, and logistic regression. These techniques usually yield accuracy ranging from 75% to 90% for similar tasks [33,34]. The improvement can be credited to the SDLC’s ensemble approach, which utilises the strengths of various deep learning models, such as the Adaptive Convolutional Neural Network (ACNN) and the Stacked Bidirectional Long Short-Term Memory (SBi-LSTM) network, to analyse intricate patterns in multi-modal data effectively. On the other hand, previous studies that have used single deep learning models or traditional methods have found slightly lower accuracies, typically ranging from 85% to 92%. By combining the ACNN and SBi-LSTM models, the SDLC can effectively incorporate various features from transcriptomic, clinical, and laboratory data, resulting in improved diagnostic performance. This comparison highlights the benefits of utilising advanced ensemble deep learning techniques, such as the SDLC, to capture complex biological signals for precise SLE diagnosis. It showcases the superiority of our approach over traditional and other contemporary methods.

Regarding clinical outcomes, the SDLC model demonstrated a significant improvement in diagnostic accuracy, achieving 99.6% accuracy, substantially higher than traditional diagnostic methods. This enhanced accuracy allows for an earlier and more precise diagnosis of Systemic Lupus Erythematosus (SLE), enabling timely intervention to prevent disease progression and reduce the risk of severe complications. By providing more accurate diagnoses, the SDLC model facilitates personalised treatment plans, leading to better patient management and improved quality of life.

### Ablation Study

Each row represents a separate experiment to assess the relative importance of various model components on overall performance. With a high precision score of 99.2% and a recall score of 99.7%, the “Full Model” experiment illustrating the full Voting Classifier model reached an accuracy of 99.6%. The accuracy dropped to 98.6% after the attention-based CNN (ACNN) component was removed from the model (“Without ACNN” trial), suggesting that ACNN favourably impacts overall performance. Similarly, the accuracy dropped to 98.8% in the “Without SBLSTM” trial, indicating that SBLSTM is crucial for enhancing the model’s predictive skills. Table 8 displays the outcomes of the Voting Classifier model’s ablation investigation.

Curiously, the accuracy dropped to 99% in the “Without Attention Mechanism” trial, suggesting that the attention mechanism is vital for collecting important aspects in classification tasks; its removal from the model did not affect the accuracy. The “Without Max Pooling Layers” experiment slightly improved accuracy to 99.3% after eliminating the Max Pooling layers from the model, indicating that these layers may not significantly influence the model’s performance in this particular scenario. The value of the Voting Classifier in utilising the diversity of different models to produce improved predictive performance was highlighted in the “Without Voting” experiment, where the accuracy dropped to 96.7% without the ensemble approach. When taken as a whole, the ablation study sheds light on how several parts of the Voting Classifier work together to help in SLE diagnosis.

The suggested approach has many benefits, such as the following:Using Robust Feature Selection to train on the most valuable data increases predictive power.Preprocessing and data integration raise the bar for input data quality and consistency.Improve your model’s efficiency via hyperparameter tuning.To increase accuracy and resilience, a Stacked deep learning classifier integrates the best features of several classifiers.A balanced dataset ensures excellent accuracy and dependability by preventing bias towards any class.The attention mechanism aids the model in zeroing in on the most relevant data to improve prediction accuracy.

Critical issues with the suggested approach include the following:Both the amount and quality of the input data have a significant impact on performance.Training and inference need a significant amount of computer resources.Learning models are more challenging to understand than simpler ones.The selection of hyperparameters may have an impact on performance.

Potential pitfalls of the model include the following:Predictions made with missing or partial data can be off.Performance may be negatively affected by input data with high noise levels.A skewed model towards the majority class can result from a class imbalance.Problems could arise if presented with datasets with different feature distributions.

## 5. Conclusions

This paper presents a new method for diagnosing SLE using deep learning techniques. Specifically, we used a Voting Classifier ensemble that included attention-based CNN (ACNN) and Stacked Bi-LSTM (SBLSTM) models. With a remarkable accuracy of 99.6% on the test dataset, our results show that the suggested method is viable for correctly categorising SLE cases. Our ablation investigation helped us determine which parts of the Voting Classifier model—ACNN, SBLSTM, attention mechanism, and max pooling layers—were most important for its overall performance. Our suggested model is a strong candidate for doctors to assist in diagnosing SLE due to its high accuracy and resilience, which could translate into earlier identification and better patient outcomes. Our method offers a practical and dependable way to analyse diverse data sources, such as gene expression profiles, clinical data, and medical images, to aid in precise diagnosis. This is achieved by utilising deep learning techniques and ensemble learning procedures.

Although this study’s findings are encouraging, there are still many opportunities for future research that could further improve the suggested method. Incorporating new data sources, like genomic data and electronic health records, could enhance the model’s predictive accuracy and offer a more thorough knowledge of the condition. Investigating more complex deep learning structures and optimisation methods might also lead to additional advancements in precision and efficacy.

## Figures and Tables

**Figure 1 diagnostics-14-01339-f001:**
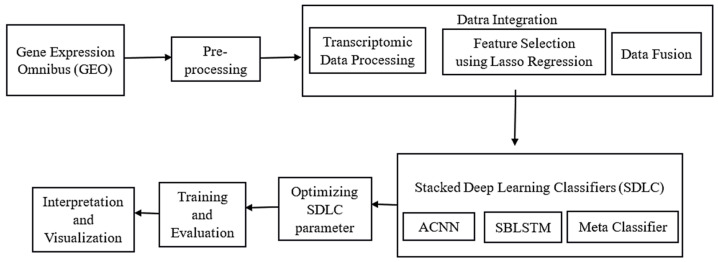
The overall processing flow of the proposed methodology for recognising SLE.

**Figure 2 diagnostics-14-01339-f002:**
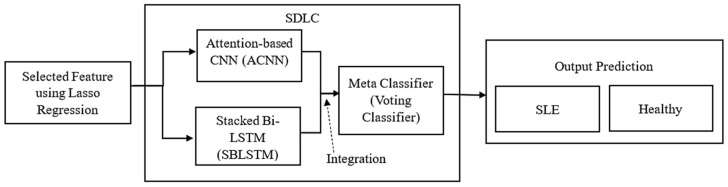
The proposed methodology Stacked Deep Learning Classifier (SDLC).

**Figure 3 diagnostics-14-01339-f003:**
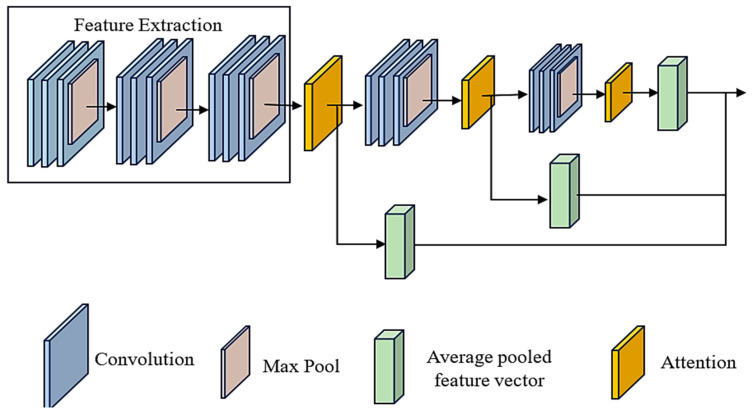
The architecture of attention-based CNN model (ACNN).

**Figure 4 diagnostics-14-01339-f004:**
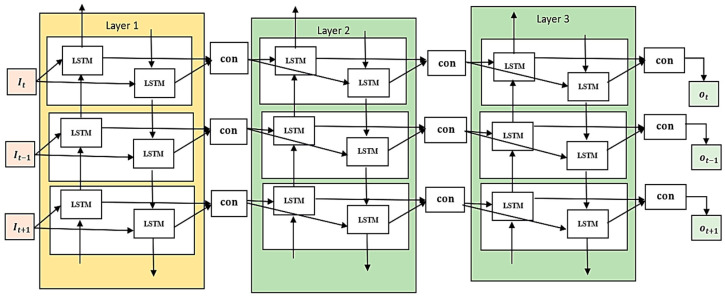
Stacked BiLSTM architecture.

**Figure 5 diagnostics-14-01339-f005:**
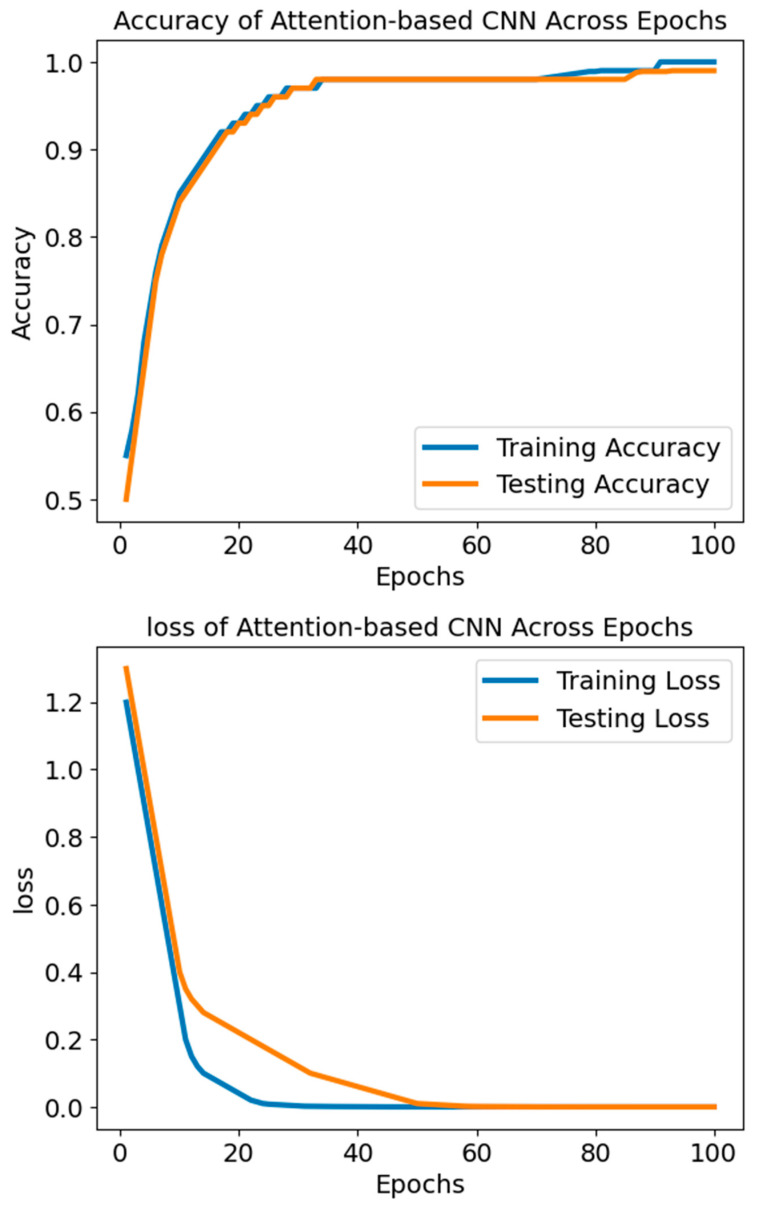
The accuracy and loss of the model ACNN.

**Figure 6 diagnostics-14-01339-f006:**
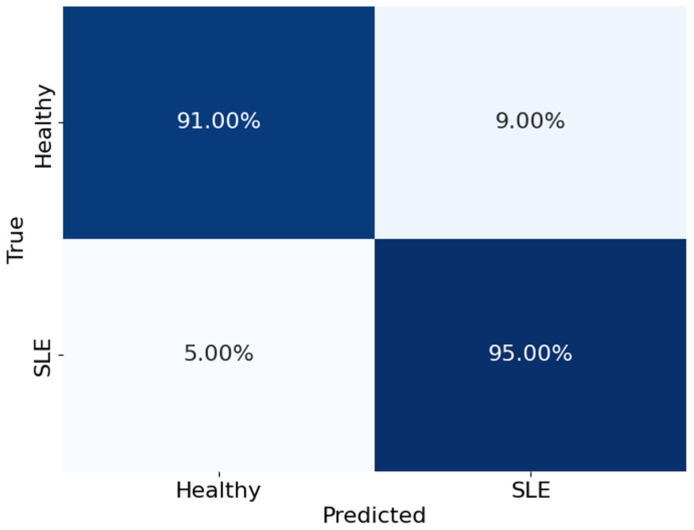
Confusion matrix of the model ACNN.

**Figure 7 diagnostics-14-01339-f007:**
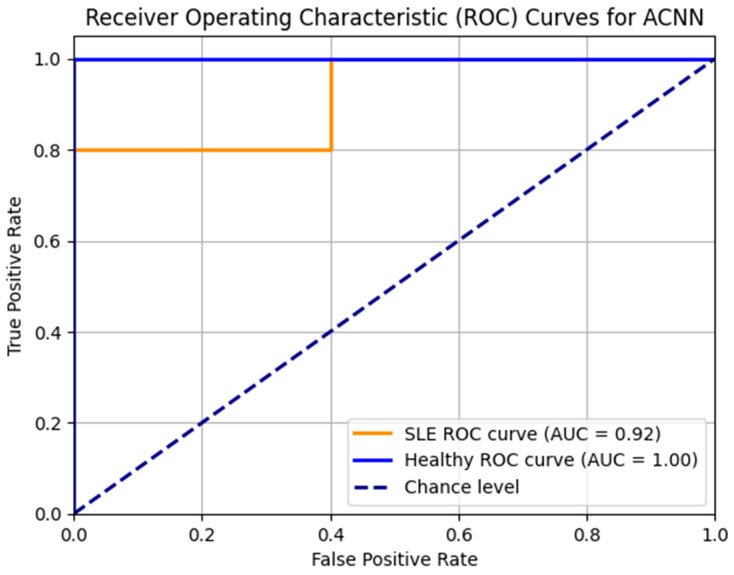
ROC_AUC curve for the ACNN model.

**Figure 8 diagnostics-14-01339-f008:**
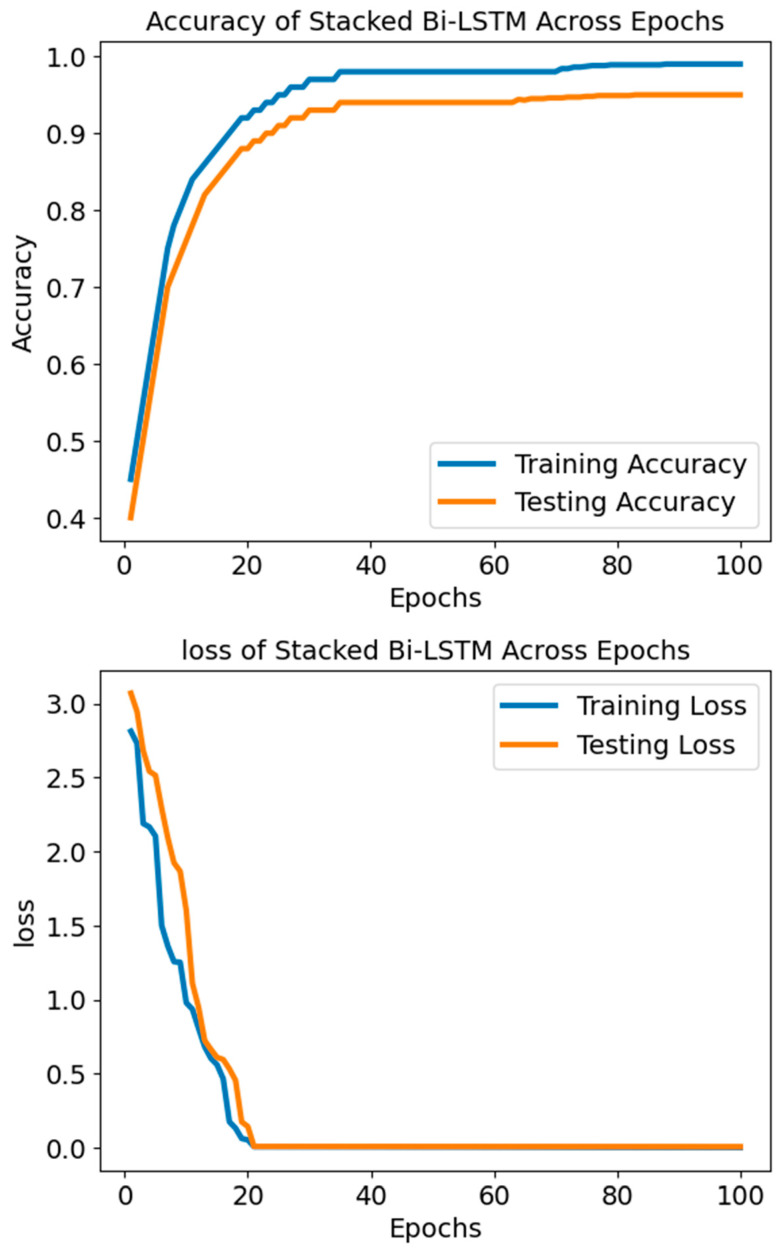
The accuracy and loss of the model SBLSTM.

**Figure 9 diagnostics-14-01339-f009:**
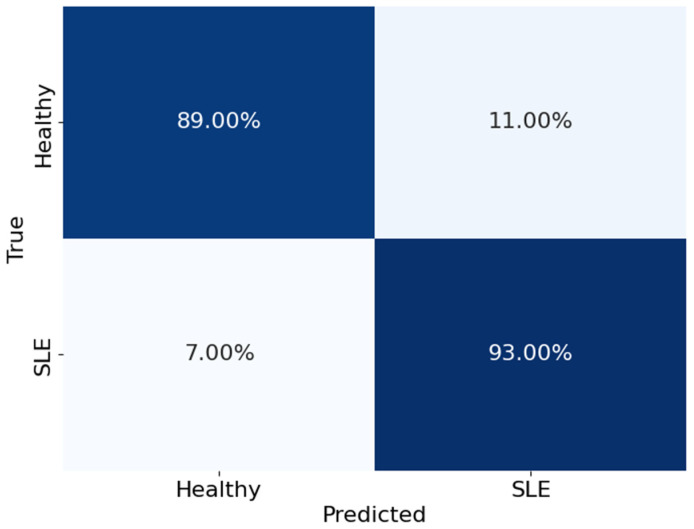
The confusion matrix of the model SBLSTM.

**Figure 10 diagnostics-14-01339-f010:**
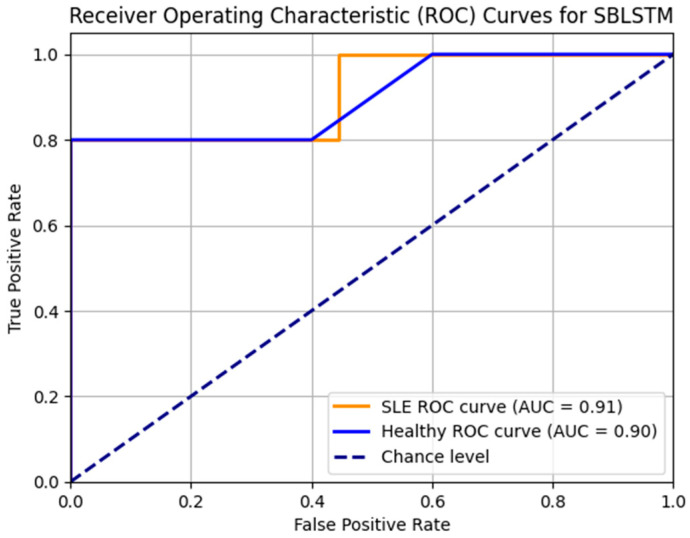
ROC_AUC curve for the model SBLSTM.

**Figure 11 diagnostics-14-01339-f011:**
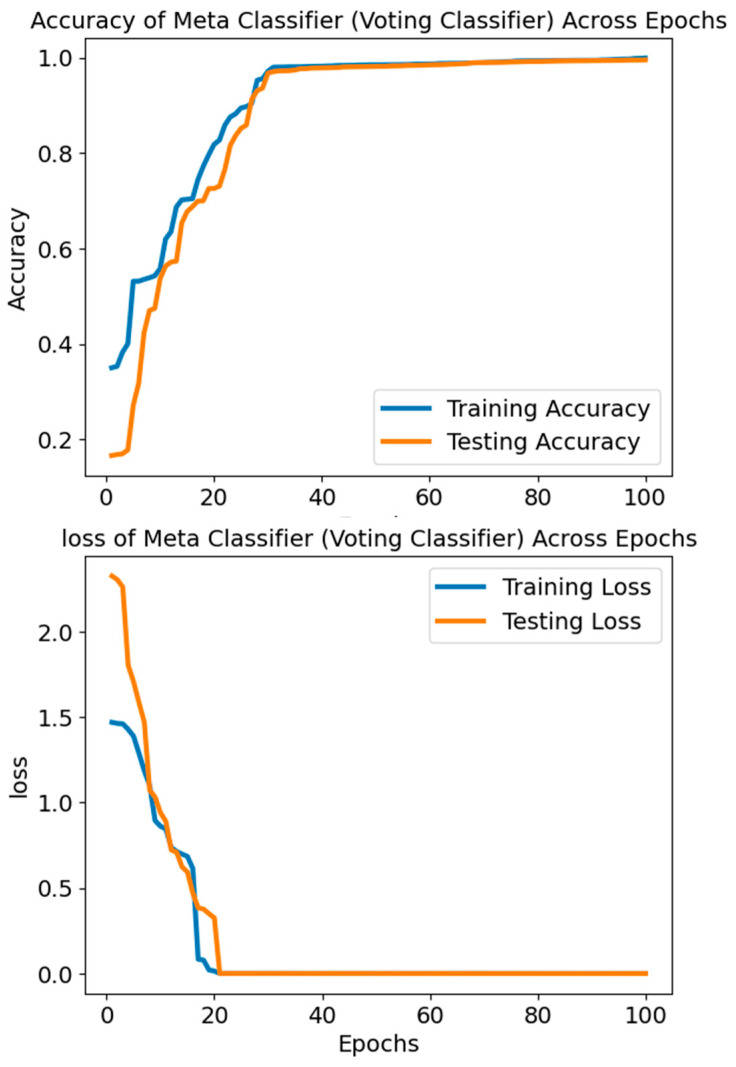
The accuracy and loss of the Voting Classifier.

**Figure 12 diagnostics-14-01339-f012:**
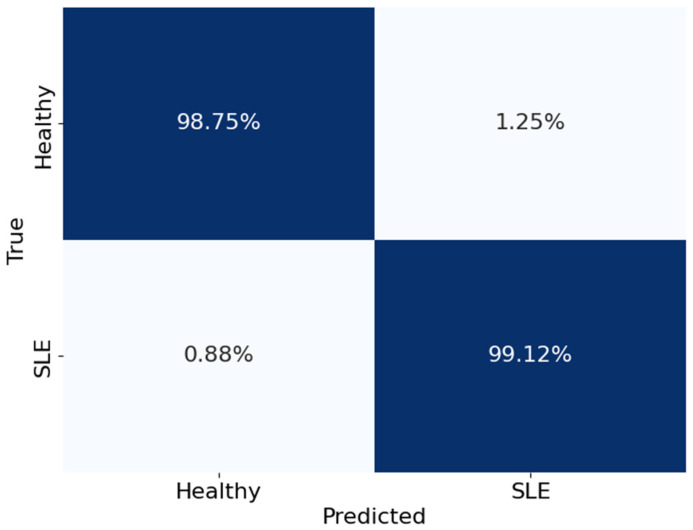
The confusion matrix of the Voting Classifier.

**Figure 13 diagnostics-14-01339-f013:**
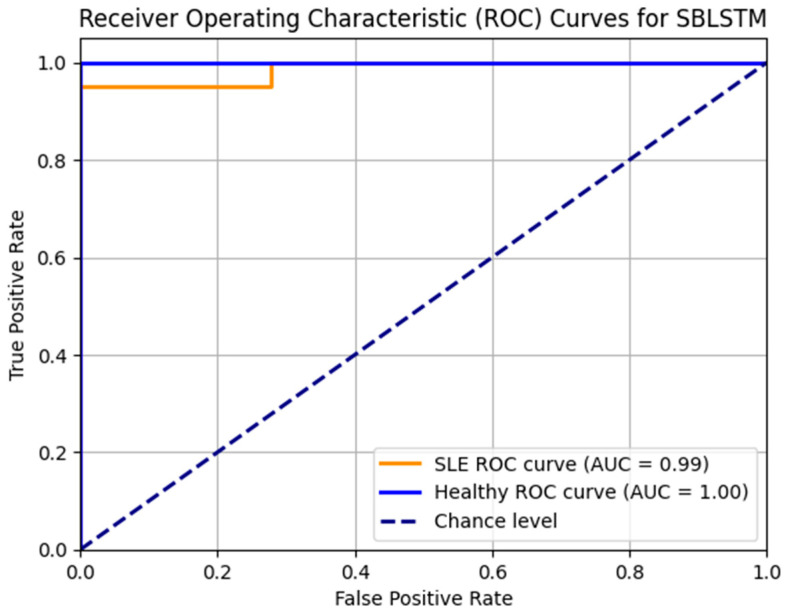
ROC_AUC curve of the model Voting Classifier.

**Table 1 diagnostics-14-01339-t001:** Data distribution from the dataset gene expression omnibus (GEO).

GEO Accession	SLE	Health	Total
GSE138458	307	23	330
GSE154851	38	32	70
GSE50635	33	16	49
GSE61635	99	30	129
GSE99967	38	17	55
GSE185047	87	0	87
GSE110685	36	17	53
GSE112087	62	58	120
GSE72509	99	18	117

GSE—Gene Expression Omnibus Series.

**Table 2 diagnostics-14-01339-t002:** ACNN model architecture with parameters.

Layer (Type)	Param #	Output Shape
input_1	[(None, 32, 32, 3)]	0
conv2d	(None, 32, 32, 32)	896
max_pooling2d	(None, 16, 16, 32)	0
conv2d_1	(None, 16, 16, 64)	18,496
max_pooling2d_1	(None, 8, 8, 64)	0
conv2d_2	(None, 8, 8, 128)	73,856
max_pooling2d_2	(None, 4, 4, 128)	0
dot_product_attention	(None, 4, 4, 128)	0
conv2d_3	(None, 4, 4, 256)	295,168
max_pooling2d_3	(None, 2, 2, 256)	0
dot_product_attention_1	(None, 2, 2, 256)	0
conv2d_4	(None, 2, 2, 512)	1,180,160
max_pooling2d_4	(None, 1, 1, 512)	0
global_average_pooling2d	(None, 512)	0
concatenate	(None, 1280)	0
dense	(None, 512)	655,872
dense_1	(None, 2)	1026
Total params:	22, 20, 474	
Trainable params:	22, 20, 474	
Non-trainable params:	0	

**Table 3 diagnostics-14-01339-t003:** Hyperparameter tuning for the attention-based CNN and SBLSTM.

Hyperparameter	Attention-Based CNN	Stacked Bi-LSTM
Number of Convolutional Layers	5	-
Number of Filters (Convolutional Layers)	[32, 64, 128, 256, 512]	-
Filter Size (Convolutional Layers)		-
Pooling Size (Max Pooling Layers)	(2, 2)	-
Dropout Rate	0.4	-
Learning Rate	0.001	0.01
Batch Size	32	64
Number of Bi-LSTM Layers	-	3
Number of Units (Bi-LSTM Layers)	-	[64, 128, 64]
Dropout Rate (Bi-LSTM Layers)	-	0.3
Recurrent Dropout Rate (Bi-LSTM Layers)	-	0.2
Activation Function (Output Layer)	Softmax	Softmax
Loss Function	Sparse Categorical Crossentropy	Sparse Categorical Crossentropy
Optimizer	Adam	Adam
Number of Epochs	100	100

**Table 4 diagnostics-14-01339-t004:** Performance evaluation of ACNN.

Metric	Value
Accuracy	0.95
Precision	0.92
Recall	0.96
F1 Score	0.94
Specificity	0.93
AUC-ROC Score	0.97

**Table 5 diagnostics-14-01339-t005:** Performance Evaluation of SBLSTM.

Metric	Value
Accuracy	0.92
Precision	0.88
Recall	0.94
F1 Score	0.91
Specificity	0.90
AUC-ROC Score	0.94

**Table 6 diagnostics-14-01339-t006:** Performance evaluation of Meta classifier.

Metric	Value
Accuracy	0.996
Precision	0.992
Recall	0.997
F1 Score	0.994
Specificity	0.99
AUC-ROC Score	0.998

**Table 7 diagnostics-14-01339-t007:** Ten-fold cross-validation of the models ACNN, SBLSTM, and Voting.

Fold	Model	Accuracy	Precision	Recall	F1 Score	Specificity	AUC-ROC
1	ACNN	0.95	0.92	0.94	0.93	0.93	0.95
	SBLSTM	0.9	0.88	0.9	0.89	0.9	0.93
	Voting	0.99	0.93	0.95	0.94	0.95	0.97
2	ACNN	0.96	0.91	0.94	0.92	0.93	0.95
	SBLSTM	0.91	0.89	0.91	0.9	0.91	0.94
	Voting	0.98	0.92	0.94	0.93	0.94	0.96
3	ACNN	0.94	0.9	0.93	0.91	0.92	0.94
	SBLSTM	0.89	0.87	0.89	0.88	0.89	0.92
	Voting	0.99	0.93	0.95	0.94	0.95	0.97
4	ACNN	0.95	0.92	0.94	0.93	0.93	0.95
	SBLSTM	0.9	0.88	0.9	0.89	0.9	0.93
	Voting	0.99	0.93	0.95	0.94	0.95	0.97
5	ACNN	0.96	0.91	0.94	0.92	0.93	0.95
	SBLSTM	0.91	0.89	0.91	0.9	0.91	0.94
	Voting	0.99	0.93	0.95	0.94	0.95	0.97
6	ACNN	0.95	0.92	0.94	0.93	0.93	0.95
	SBLSTM	0.9	0.88	0.9	0.89	0.9	0.93
	Voting	0.99	0.93	0.95	0.94	0.95	0.97
7	ACNN	0.96	0.91	0.94	0.92	0.93	0.95
	SBLSTM	0.91	0.89	0.91	0.9	0.91	0.94
	Voting	0.99	0.93	0.95	0.94	0.95	0.97
8	ACNN	0.94	0.9	0.93	0.91	0.92	0.94
	SBLSTM	0.89	0.87	0.89	0.88	0.89	0.92
	Voting	0.99	0.93	0.95	0.94	0.95	0.97
9	ACNN	0.95	0.92	0.94	0.93	0.93	0.95
	SBLSTM	0.9	0.88	0.9	0.89	0.9	0.93
	Voting	0.99	0.93	0.95	0.94	0.95	0.97
10	ACNN	0.95	0.92	0.94	0.93	0.93	0.95
	SBLSTM	0.9	0.88	0.9	0.89	0.9	0.93
	Voting	0.99	0.93	0.95	0.94	0.95	0.97

**Table 8 diagnostics-14-01339-t008:** Ablation study for Voting Classifier.

Experiment	Accuracy (%)	Precision (%)	Recall (%)
Full Model	99.6	99.2	99.7
Without ACNN	98.6	98.3	99
Without SBLSTM	98.8	98.4	99.2
Without Attention Mechanism	99	98.6	99.5
Without Max Pooling Layers	99.3	98.9	99.6
Without Voting	96.7	96.2	97.5

## Data Availability

Data will be made available on request.

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
