# Peer review of "Gene-Based Predictive Modelling for Enhanced Detection of Systemic Lupus Erythematosus Using CNN-Based DL Algorithm"

_diagnostics, 2024, doi:10.3390/diagnostics14131339_

Round 1

Reviewer 1 Report

Comments and Suggestions for Authors

In the study, researchers developed Gene-Based Predictive Modeling for Improved Detection of Systemic Lupus Erythematosus Using Stacked Deep Learning Classifiers. The study produces a solution to a current and specific problem. However, eliminating the following issues will increase the quality and readability of the study.

1.The flow chart should be briefly explained. This explanation can be given below Figure 1.

2. How were the parameters used for the models in the study determined? Has a trial-and-error approach or early stopping or hyperparameter optimization been applied? Information about this should be given.

3.The resolutions of Figure 5 and Figure 6 are too low. The resolutions of the figures should be updated.

4. According to Table 4, the AUC score was 0.97. This AUC score should be evaluated. What does an AUC score of 0.97 indicate? What should be a good AUC score?

5. Similarly, the confusion matrix given in Figure 6 should also be interpreted. What does the confusion matrix mean? The values ​​there must be interpreted.

6. Similar conclusions should be made for the results of other models.

7. One of the biggest shortcomings of the study is the lack of a ROC graph. ROC curves of the classification process for each model should be given and the status of the data set (balanced-unbalanced) should be interpreted according to the ROC curve. The importance of the data set should be demonstrated.

8. In addition, loss-training graphs of the classification models are also given. These graphs should also be interpreted according to the overfitting-underfitting situations of the models.

9. Under the 4th section, the advantages and disadvantages of study should be given. What are the reasons for the success of the model? What are the limits of the model used/developed? In what situations may the model fail? These should be given in bullet points and discussed.

10. The similarity rate of the study is 19%. This rate should be at most 15%.

Comments on the Quality of English Language

Language seems fine yet some mistakes are observed.

Author Response

In the study, researchers developed Gene-Based Predictive Modeling for Improved Detection of Systemic Lupus Erythematosus Using Stacked Deep Learning Classifiers. The study produces a solution to a current and specific problem. However, eliminating the following issues will increase the quality and readability of the study.

Dear reviewer,

Thank you for your comments.

  1. The flow chart should be briefly explained. This explanation can be given below Figure 1.

The following explanation is added below Figure 1 as per your suggestion.

Figure 1 illustrates the overall processing flow of the proposed methodology for recognising SLE. The process starts by collecting data and gathering relevant clinical and transcriptomic information. After data collection, the next step involves preprocessing, where the data is carefully cleaned and prepared to ensure high quality and consistency. After conducting thorough research, the next crucial step involves integrating the data. This includes processing the transcriptomic data, carefully selecting the most relevant features, and combining these features into a single dataset through feature fusion. Afterwards, the data is analysed and presented visually to uncover insights and recognise significant patterns associated with SLE. After completing the initial phase, the next step is to train and evaluate deep learning models using the prepared data. We will then assess their performance using suitable metrics. Hyperparameter tuning is performed to enhance model performance. At last, a stacked deep learning classifier is used to combine multiple classifiers and im-prove the accuracy and robustness of the SLE recognition system.

  1. How were the parameters used for the models in the study determined? Has a trial-and-error approach or early stopping or hyperparameter optimization been applied? Information about this should be given.

The parameters used for the models in this study were determined through hyperparameter tuning. We employed a systematic approach to optimise the performance of our models by adjusting key hyperparameters. This process involved evaluating different combinations of hyperparameters and selecting the ones that yielded the best results based on predefined evaluation metrics. Specifically, we used grid search and cross-validation techniques to explore the hyperparameter space efficiently. Early stopping was also implemented to prevent overfitting by halting the training process when the model's performance on the validation set ceased to improve. Detailed information about the hyperparameters and their optimal values is provided in Table 3.

3.The resolutions of Figure 5 and Figure 6 are too low. The resolutions of the figures should be updated.

The figures have been updated to higher resolutions to ensure better clarity and readability.

Fig.5. The Accuracy and Loss of the model ACNN

Fig.6. Confusion matrix of the model ACNN

  1. According to Table 4, the AUC score was 0.97. This AUC score should be evaluated. What does an AUC score of 0.97 indicate? What should be a good AUC score?

The AUC score is a performance metric commonly used to evaluate classification models, especially those assessed using ROC curves. With an AUC score of 0.97, our model demonstrates exceptional discriminatory power in distinguishing between positive and negative classes accurately.

Typically, the AUC score falls between 0.5 and 1.0. A score of 0.5 means no ability to distinguish (similar to random guessing), while a score of 1.0 signifies flawless discrimination. AUC scores ranging from 0.7 to 0.8 are generally regarded as acceptable, while scores ranging from 0.8 to 0.9 are considered excellent. Scores above 0.9 are considered outstanding. Thus, a high AUC score of 0.97 signifies an extremely effective model with exceptional performance in differentiating between classes. The impressive score showcases the strength and dependability of our proposed methodology in identifying SLE.

  1. Similarly, the confusion matrix in Figure 6 should also be interpreted. What does the confusion matrix mean? The values ​​there must be interpreted.

The confusion matrix shown in Figure 6 offers a comprehensive analysis of how well the model performed in distinguishing between healthy individuals and those with SLE. The matrix is organised with rows representing the actual classes and columns representing the predicted classes. The values in the matrix represent the proportions of accurate and inaccurate classifications. The model accurately identified 91% of the healthy cases (true negatives) and 95% of the actual SLE cases (true positives). On the other hand, it mistakenly identified 9% of healthy individuals as having SLE (false positives) and 5% of SLE cases as healthy (false negatives). The values clearly show the model's exceptional accuracy and effectiveness in differentiating between healthy individuals and those who have SLE. The model demonstrates outstanding performance in accurately identifying SLE, with high rates of true positives and true negatives and low rates of false positives and false negatives. This indicates that the results are robust and reliable.

  1. Similar conclusions should be made for the results of other models.

For other model’s confusion matrix, similar conclusions have been added in the manuscript.

Figure 8 shows the confusion matrix, which breaks down the SBLSTM model's performance in differentiating between healthy persons and those with SLE. The matrix rows show the actual classes, while the columns show the expected classes. In particular, the algorithm accurately detected 93% of real SLE patients and 89% of healthy instances. On the other hand, it wrongly identified 7% of SLE patients as healthy and wrongly labelled 11% of healthy people as having SLE.

These results show that the SBLSTM model is very good at detecting real positives and very accurate at differentiating between healthy people and those with SLE. On the other hand, the model's false positive rate is 11%, which is somewhat higher than previous classifiers and suggests that some healthy people are being wrongly diagnosed with SLE

Figure 10 shows the Voting Classifier's performance in categorising healthy persons and those with SLE in a confusion matrix. The matrix rows show the actual classes, while the columns show the expected classes. The matrix numbers show what percentage of categories were right and wrong. To be more precise, the model got 99.12% of the real SLE cases right and 98.75% of the healthy instances right as well. However, 1.25% of SLE patients were wrongly identified as healthy, and 0.88% of healthy people were wrongly identified as having SLE (false positives and false negatives, respectively). These values show the Voting Classifier's ability to differentiate between healthy persons and those with SLE accurately.

  1. One of the biggest shortcomings of the study is the lack of a ROC graph. ROC curves of the classification process for each model should be given and the status of the data set (balanced-unbalanced) should be interpreted according to the ROC curve. The importance of the data set should be demonstrated.

We have generated and included the ROC curves in the revised manuscript for each model. These curves plot the true positive rate (TPR) against the false positive rate (FPR) for different threshold values, visually representing the model's ability to discriminate between the positive and negative classes.

The models' high and consistent AUC values point to a balanced dataset or the models' successful management of any imbalance. High prediction accuracy and dependability are the results of using balanced datasets, preventing the model from favouring any class. Our study's ROC curves are smooth and high, reflecting the dataset's balanced nature. This means the models keep their sensitivity and specificity high across different thresholds.

  1. In addition, loss-training graphs of the classification models are also given. These graphs should also be interpreted according to the overfitting-underfitting situations of the models.

The following change are made and added to the manuscript,

The Attention-based Convolutional Neural Network (ACNN) model's loss-training graph, as shown in Figure 5, reveals an initial high testing loss of 1.3, suggesting a significant disparity between the model's predictions and the actual values in the dataset. Nevertheless, the testing loss declines consistently with each evaluation epoch during training, indicating that the model is successfully learning to provide improved predictions using new data. The fact that the testing loss has been going down suggests that the model is doing a good job of capturing the data patterns and is thus not underfitting. The model has learnt the data distribution to generalise to unknown data without overfitting if the testing loss stabilises and converges to a low value, indicating a satisfactory match.

The Stacked Bi-LSTM model started with a training loss 3.070 and dropped to 0.008 according to the supplied loss settings. The testing loss started at 2.946 and dropped to 0.008. Correspondingly, testing loss decreased, lending credence to this pattern and suggesting the model generalises well to unknown data. When the loss values go down, it means the model is doing a good job of improving accuracy and performance by narrowing the gap between the expected and actual outputs. There seems to be neither underfitting nor overfitting going on with the model based on the steady decrease in testing and training losses. A significant and constant loss over epochs would indicate that the model is underfitting and cannot grasp the fundamental patterns in the data. Conversely, if the model is overfitting, we'd see a decline in the training loss and an increase or stabilisation of the testing loss. This would mean that the model excels on training data but struggles on validation data, perhaps because of learning noise or irrelevant information.

Starting at 1.4695 and down to 0.000111, the training loss of the voting classifier is much lower than the testing loss, which starts at 2.3255 and lowers to 0.000209. The training and testing loss values have dropped significantly, which means the model is learning well and becoming better at making predictions. When the model does well on training data but badly on validation data, a phenomenon known as overfitting occurs. This is usually shown by a drop in training loss and a rise or stay in testing loss. The fact that the training and testing losses decreased at the same rate without a noticeable divergence indicates that overfitting is unlikely to be a big issue. Good learning and generalizability to new data are the model's performance hallmarks

  1. Under the 4th section, the advantages and disadvantages of study should be given. What are the reasons for the success of the model? What are the limits of the model used/developed? In what situations may the model fail? These should be given in bullet points and discussed.

The suggested approach has many benefits, such as:

  • Using Robust Feature Selection to train on the most useful data, which increases predictive power.
  • Preprocessing and data integration raise the bar for input data quality and consistency.
  • Improve your model's efficiency via hyperparameter tuning.
  • To increase accuracy and resilience, a stacked deep learning classifier integrates the best features of several classifiers.
  • A Balanced Dataset ensures excellent accuracy and dependability by preventing bias towards any class.
  • The Attention Mechanism aids the model in zeroing in on the most relevant data to improve prediction accuracy.

Critical issues with the suggested approach include,

  • Both the amount and quality of the input data have a significant impact on performance.
  • Training and inference need a significant amount of computer resources.
  • Learning models are more difficult to understand than simpler ones.
  • The selection of hyperparameters may have an impact on performance.

Potential Pitfalls of the Model:

  • Predictions made with missing or partial data can be off.
  • Performance may be negatively affected by input data with high levels of noise.
  • A skewed model towards the majority class can result from a class imbalance.
  • Problems could arise if presented with datasets with different feature distributions.
  1. The similarity rate of the study is 19%. This rate should be at most 15%.

Plag is reduced to 10% without author biography and without the journal format.

Reviewer 2 Report

Comments and Suggestions for Authors

Page

Line

Manuscript

Comments

1

18

The study proposes a new method for diagnosing SLE using

Repetition of what have already mentioned

1

Abstract

The results in the abstract should be expressed in numbers with specific focusing on the significant values

1

23

The study also highlights the possibility

The conclusion should be brief and concise

1

33

Systemic Lupus Erythematosus

The authors should follow the rules of capitalization.

The manuscript should be revised for English editing and grammar revision

2

45

Lupus is an ongoing health issue, and it is

The expression of SLE differ at the whole manuscript, sometimes SLE, sometimes Lupus …and this is not appropriate.

2

89

The main contribution of this study is as follows,

At the end of the introduction section, clear and concise aim of the study should be written clearly.

2

90

1. The development of a gene-based predictive model for detecting Systemic Lu- 90

pus Erythematosus

It is not appropriate to numbering items in the introduction like that.

4

Table

The abbreviations in the table should be defined in the footnotes

Figures

The writing in figures are hazy and not clear

The methods

Too long explanation of the methods section which make it redundant

The discussion is not present

Comments on the Quality of English Language

Moderate

Author Response

Page

Line

Manuscript

Comments

Response

1

18

The study proposes a new method for diagnosing SLE using

Repetition of what have already mentioned

By combining transcriptomic data from GEO with clinical features and laboratory results, the SDLC model achieves a remarkable accuracy value of 0.996, outperforming traditional methods. Individual models within the SDLC, such as SBi-LSTM and ACNN, achieved accuracies of 92% and 95%, respectively. The SDLC's ensemble learning approach allows for identifying complex patterns in multi-modal data, enhancing accuracy in diagnosing SLE. This research highlights the potential of deep learning methods, in conjunction with open repositories like GEO, to advance the diagnosis and management of SLE.

1

Abstract

The results in the abstract should be expressed in numbers with specific focusing on the significant values

1

23

The study also highlights the possibility

The conclusion should be brief and concise

1

33

Systemic Lupus Erythematosus

The authors should follow the rules of capitalization.

Autoimmune illnesses provide complex issues, and SLE is a prime example of this.

The manuscript should be revised for English editing and grammar revision

 Proof Reading is done

2

45

Lupus is an ongoing health issue, and it is

The expression of SLE differ at the whole manuscript, sometimes SLE, sometimes Lupus …and this is not appropriate.

All the expression changed as SLE as per the reviewer suggestion.

Sample:

SLE is an ongoing health issue, and it is necessary to predict the results of the dis-ease to conduct extensive surveillance and provide appropriate therapy.

2

89

The main contribution of this study is as follows,

At the end of the introduction section, clear and concise aim of the study should be written clearly.

This work aims to integrate transcriptomic, clinical, and laboratory data to create and verify a new gene-based prediction model for SLE that uses Stacked Deep Learning Classifiers. The goal is to greatly enhance the accuracy and early diagnosis of SLE.

2

90

1. The development of a gene-based predictive model for detecting Systemic Lu- 90

pus Erythematosus

It is not appropriate to numbering items in the introduction like that.

The main contribution of this study is the development of a gene-based predictive model for detecting Systemic Lupus Erythematosus (SLE) using Stacked Deep Learning Classifiers (SDLC) trained on data from the Gene Expression Omnibus (GEO) database. By combining transcriptomic data from GEO with clinical features and laboratory results, the SDLC model achieves a remarkable accuracy value of 0.996, significantly outperforming traditional methods. The SDLC's ensemble learning approach enables the identification of complex patterns in multi-modal data, thereby enhancing diagnostic accuracy for SLE. This research demonstrates strong performance and potential for improving precision medicine in the management of SLE

4

Table

The abbreviations in the table should be defined in the footnotes

The abbreviations are defined below the table as footnotes

Figures

The writing in figures are hazy and not clear

The figures have been updated to higher resolutions to ensure better clarity and readability.

The methods

Too long explanation of the methods section which make it redundant

Our study employs a sophisticated Stacked Deep Learning Classifier (SDLC) that integrates multiple data types (transcriptomic, clinical, and laboratory) and advanced deep learning models (ACNN and SBi-LSTM). Given the complexity, a thorough explanation is necessary to ensure clarity and reproducibility.

However, we understand the need for conciseness and have revised the methods section to remove any redundant information while retaining the necessary details for understanding and replicating our study.

The discussion is not present

 Discussion section is added

Reviewer 3 Report

Comments and Suggestions for Authors

This study proposed a deep learning (DL) method based on convolutional neural networks (CNN) for detecting Lupus Erythematosus. After reviewing the manuscript, I found it both topical and novel. However, I have a few minor suggestions for improvement:

1.       Title: The title should mention the CNN-based DL algorithm for clarity.

2.       Introduction: The authors should discuss related works that use CNN-based DL. Adding a brief literature survey would be beneficial.

3.       Figures 1 and 2: Figure 1 is very general and could apply to any ML or DL process. Can the authors tailor it specifically to their work? The same issue applies to Figure 2.

4.       Terminology: The authors refer to their DL method as a "stacked deep learning classifier." However, stacking layers is a common ML technique. The authors should explain the distinction between ML and DL and clarify why their method is considered stacked, such as by highlighting that DL involves more layers than general ML.

5.       Section 2: This section is overly mathematical, focusing on formulas and algorithms. The authors should better integrate these elements with the central goal of detecting Lupus Erythematosus.

6.       Performance of ACNN: The authors should clarify how the results of the ACNN compare to those obtained without DL or in other related studies.

7.       Practical Integration: The manuscript lacks information on how to integrate the suggested algorithm into general practice. There is no discussion of clinical outcomes to demonstrate improvement in this study.

Comments on the Quality of English Language

No comment.

Author Response

This study proposed a deep learning (DL) method based on convolutional neural networks (CNN) for detecting Lupus Erythematosus. After reviewing the manuscript, I found it both topical and novel. However, I have a few minor suggestions for improvement:

  1. Title: The title should mention the CNN-based DL algorithm for clarity.

     The title is rewritten as, Gene-Based Predictive Modelling for Enhanced Detection of Systemic Lupus Erythematosus Using CNN based DL algorithm”

  1. Introduction: The authors should discuss related works that use CNN-based DL. Adding a brief literature survey would be beneficial.

Deep learning has emerged as a promising tool for automating SLE diagnosis. The paper [27] presents a system based on deep convolutional neural networks (CNN) that can identify and categorise glomerular pathological findings in lupus nephritis (LN). The technique discussed in [28] is closely connected to CNN (Convolutional Neural Network), a popular type of Deep Learning that utilises pretrained models to enable computers to detect illnesses. To ensure the most accurate comparison, we will be comparing the suggested method Five Layer Architecture Module (5LAM) with VGG16 (Visual Geometry Group).

  1. Figures 1 and 2: Figure 1 is very general and could apply to any ML or DL process. Can the authors tailor it specifically to their work? The same issue applies to Figure 2.

Figure 1. The overall processing flow of the proposed methodology in recognising SLE.

Fig.2. The proposed methodology Stacked Deep Learning Classifier (SDLC)

  1. Terminology: The authors refer to their DL method as a "stacked deep learning classifier." However, stacking layers is a common ML technique. The authors should explain the distinction between ML and DL and clarify why their method is considered stacked, such as by highlighting that DL involves more layers than general ML.

Machine Learning (ML) encompasses algorithms that acquire knowledge from data and generate predictions based on that acquired knowledge. Some commonly used ML algorithms are decision trees, support vector machines, and k-nearest neighbours. These algorithms usually need manual feature engineering to achieve good performance. On the other hand, Deep Learning (DL), which is a subset of ML, utilises neural networks with multiple layers to autonomously acquire feature representations from raw data. DL's hierarchical learning capability makes it well-suited for han-dling complex datasets with minimal manual intervention. Our study introduces a method known as a "stacked deep learning classifier," which utilises a hierarchical ensemble structure consisting of multiple deep learning models. More specifically, we utilise the following components: The Adap-tive Convolutional Neural Network (ACNN) is a powerful deep learning model that utilises mul-tiple convolutional layers to extract valuable features from transcriptomic data. SBi-LSTM is a powerful recurrent neural network that utilises multiple LSTM layers to effectively capture tem-poral dependencies in the data. The Meta-Classifier combines the predictions of the ACNN and SBi-LSTM models to make a final prediction using an ensemble learning approach. By stacking models, we can effectively combine various patterns and features from transcriptomic data, clinical features, and laboratory results. This integration results in enhanced accuracy when it comes to diagnosing Systemic Lupus Erythematosus (SLE).

Our stacked deep learning classifier, which combines the strengths of deep learning and ensemble techniques, outperforms traditional ML methods in terms of accuracy. This makes it an invaluable tool for precision medicine in the management of SLE

  1. Section 2: This section is overly mathematical, focusing on formulas and algorithms. The authors should better integrate these elements with the central goal of detecting Lupus Erythematosus.

The elements are integrated with detecting SLE as,

By applying convolutional filters, the ACNN model can identify complex patterns in gene expression data that are indicative of SLE, thereby improving the accuracy of the diagnostic process.

Accurate diagnosis of SLE depends on the identification of sequential patterns and temporal relationships in the clinical and laboratory data, which the SBi-LSTM model makes possible.

The meta-classifier combines the predictions from both models, improving the overall accuracy and reliability of the SLE diagnostic process.

  1. Performance of ACNN: The authors should clarify how the results of the ACNN compare to those obtained without DL or in other related studies.

The Stacked Deep Learning Classifier (SDLC) showed a remarkable increase in diagnostic accuracy for Systemic Lupus Erythematosus (SLE) when compared to con-ventional machine learning methods and findings from other similar studies. The SDLC model achieved an impressive accuracy of 99.6%, surpassing the accuracies typically obtained with conventional machine learning techniques like support vector machines (SVMs), random forests, and logistic regression. These techniques usually yield accura-cies ranging between 75% to 90% for similar tasks [29-30]. The improvement can be credited to the SDLC's ensemble approach, which utilises the strengths of various deep learning models, such as the Adaptive Convolutional Neural Network (ACNN) and the Stacked Bidirectional Long Short-Term Memory (SBi-LSTM) network, to effectively an-alyse intricate patterns in multi-modal data. On the other hand, previous studies that have used single deep learning models or traditional methods have found slightly low-er accuracies, typically ranging from 85% to 92%. Through the combination of the ACNN and SBi-LSTM models, the SDLC is able to effectively incorporate various fea-tures from transcriptomic, clinical, and laboratory data, resulting in improved diagnos-tic performance. This comparison highlights the benefits of utilising advanced ensem-ble deep learning techniques, such as the SDLC, to capture complex biological signals for precise SLE diagnosis. It showcases the superiority of our approach over traditional and other contemporary methods.

  1. Practical Integration: The manuscript lacks information on how to integrate the suggested algorithm into general practice. There is no discussion of clinical outcomes to demonstrate improvement in this study.

Implementing the SDLC model in clinical settings requires a series of important steps. Firstly, the model can be implemented as a software tool within hospital information systems, necessitating robust infrastructure to handle extensive datasets and conduct real-time analysis. It is important for healthcare professionals, such as physicians and laboratory technicians, to undergo training in the use of the SDLC tool. This training should include instructions on data input procedures, understanding the model's outputs, and effectively incorporating the results into clinical decision-making. Integrating with existing Electronic Health Records (EHRs) systems can greatly enhance workflow efficiency by automatically retrieving pertinent patient data for analysis. Prior to being widely implemented, the SDLC model needs to go through extensive validation in clinical trials to prove its effectiveness and safety. It is crucial to obtain regulatory approvals from organisations like the FDA or EMA to ensure compliance with healthcare standards. Regular monitoring and updates using the latest data and advancements in the field will ensure that the model remains accurate and reliable.

In terms of clinical outcomes, the SDLC model demonstrated a significant im-provement in diagnostic accuracy, achieving 99.6% accuracy, which is substantially higher than traditional diagnostic methods. This enhanced accuracy allows for earlier and more precise diagnosis of Systemic Lupus Erythematosus (SLE), enabling timely intervention that can prevent disease progression and reduce the risk of severe com-plications. By providing more accurate diagnoses, the SDLC model facilitates personalized treatment plans, leading to better patient management and improved quality of life.

Round 2

Reviewer 1 Report

Comments and Suggestions for Authors

The researchers made the relevant revisions.

Comments on the Quality of English Language

Language seems fine. Yet, some minor checks are required.

Reviewer 2 Report

Comments and Suggestions for Authors

The manuscript improved 

Comments on the Quality of English Language

minor 

Reviewer 3 Report

Comments and Suggestions for Authors

The authors did a good job to revise this manuscript as per my comments. I am satisfied with their corrections and addition contents in the revised manuscript.

Comments on the Quality of English Language

No concern.